# Tandem Anchoring:
# a Multiword Anchor Approach for Interactive Topic Modeling

## Abstract

Interactive topic models are powerful tools for those seeking to understand large collections of text. However, existing sampling-based interactive topic modeling approaches scale poorly to large data sets. Anchor methods, which use a single word to uniquely identify a topic, offer the speed needed for interactive work but lack both a mechanism to inject prior knowledge and lack the intuitive semantics needed for user-facing applications. We propose combinations of words as anchors, going beyond existing single word anchor algorithms—an approach we call "Tandem Anchors". We begin with a synthetic investigation of this approach then apply the approach to interactive topic modeling in a user study and compare it to interactive and non-interactive approaches. Tandem anchors are faster and more intuitive than existing interactive approaches.

Topic models distill large collections of text into topics, giving a high-level summary of the thematic structure of the data without manual annotation. In addition to facilitating discovery of topical trends (Gardner et al., 2010), topic modeling is used for a wide variety of problems including document classification (Rubin et al., 2012), information retrieval (Wei and Croft, 2006), author identification (Rosen-Zvi et al., 2004), and sentiment analysis (Titov and McDonald, 2008). However, the most compelling use of topic models is to help users understand large datasets (Chuang et al., 2012).

Interactive topic modeling (Hu et al., 2014) allows non-experts to refine automatically generated topics, making topic models less of a "take it or leave it" proposition. Including humans input during training improves the quality of the model and allows users to guide topics in a specific way, custom tailoring the model for a specific downstream task or analysis.

The downside is that interactive topic modeling is slow—algorithms typically scale with the size of the corpus—and requires non-intuitive information from the user in the form of must-link and cannot-link constraints (Andrzejewski et al., 2009). We address these shortcomings of interactive topic modeling by using an interactive version of the *anchor words* algorithm for topic models.

The anchor algorithm (Arora et al., 2013) is an alternative topic modeling algorithm which scales with the number of unique word types in the data rather than the number of documents or tokens (Section 1). This makes the anchor algorithm fast enough for interactive use, even in web-scale document collections.

A drawback of the anchor method is that anchor words—words that have high probability of being in a *single* topic—are not intuitive. We extend the anchor algorithm to use multiple anchor words in tandem (Section 2). Tandem anchors not only improve interactive refinement, but also make the underlying anchor-based method more intuitive.

For interactive topic modeling, tandem anchors produce higher quality topics than single word anchors (Section 3). Tandem anchors provide a framework for fast interactive topic modeling: users improve and refine an existing model through multiword anchors (Section 4). Compared to existing methods such as Interactive Topic Models (Hu et al., 2014), our method is much faster.

## 1   Vanilla Anchor Algorithm

The anchor algorithm computes the topic matrix $\boldsymbol{A}$, where $\boldsymbol{A}_{v,k}$ is the conditional probability of observing word $v$ given topic $k$, e.g., the probability of seeing the word "lens" given the camera topic in a corpus of Amazon product reviews. Arora et al. (2012) finds these probabilities by assuming that every topic contains at least one 'anchor' word which has a non-zero probability only in that topic. Anchor words make computing the topic matrix $\boldsymbol{A}$ tractable because the occurrence pattern of the anchor word mirrors the occurrence pattern of the topic itself.

To recover the topic matrix $\boldsymbol{A}$ using anchor words, we first compute a $V \times V$ cooccurrence matrix $\boldsymbol{Q}$, where $\boldsymbol{Q}_{i,j}$ is the conditional probability $p(w_j \,|\, w_i)$ of seeing word type $w_j$ after having seen $w_i$ in the same document. A form of the Gram-Schmidt process on $\boldsymbol{Q}$ finds anchor words $\{g_1 \ldots g_k\}$ (Arora et al., 2013).

Once we have the set of anchor words, we can compute the probability of a topic given a word (the inverse of the conditioning in $\boldsymbol{A}$). This coefficient matrix $\boldsymbol{C}$ is defined row-wise for each word $i$

$$\boldsymbol{C}_{i,\cdot}^* = \operatorname*{argmin}_{C_{i,\cdot}} D_{KL}\left(\boldsymbol{Q}_{i,\cdot} \,\Big\|\, \sum_{k=1}^{K} \boldsymbol{C}_{i,k}\boldsymbol{Q}_{g_k,\cdot}\right),$$
(1)

which gives the best reconstruction (based on Kullback-Leibler divergence $D_{KL}$) of non-anchor words given anchor words' conditional probabilities. For example, in our product review data, a word such as "battery" is a convex combination of the anchor words' contexts ($\boldsymbol{Q}_{g_k,\cdot}$) such as "camera", "phone", and "car". Solving each row of $\boldsymbol{C}$ is fast and is embarrassingly parallel. Finally, we apply Bayes' rule to recover the topic matrix $\boldsymbol{A}$ from the coefficient matrix $\boldsymbol{C}$.

The anchor algorithm can be orders of magnitude faster than probabilistic inference. The construction of $\boldsymbol{Q}$ requires only a single pass over the data and can be pre-computed for interactive use-cases. Once $\boldsymbol{Q}$ is constructed, topic inference scales with the size of $\boldsymbol{Q}$ which, in turn, depends on the square of the vocabulary size $V$. In contrast, traditional topic model inference typically requires multiple passes over the entire data. Techniques such as Online LDA (Hoffman et al., 2010) or Stochastic Variation Inference (Hoffman et al., 2013) could improve this to a single pass over the

| Anchor | Top Words in Topics |
|---|---|
| backpack | backpack camera lens bag room carry fit cameras equipment comfortable |
| camera | camera lens pictures canon digital lenses batteries filter mm photos |
| bag | bag camera diaper lens bags genie smell room diapers odor |

Table 1: Three separate attempts to construct a topic concerning camera bags in Amazon product reviews with single word anchors. The term "backpack" is a good anchor because it uniquely identifies the topic. However, both "camera" and "bag" are poor anchors for this topic.

entire data. However, even if such techniques were to be adapted to incorporate human guidance, a single pass is not tractable for interactive use.

## 2   Tandem Anchor Extension

Single word anchors can be opaque to users. For an example of bewildering anchor words, consider a camera bag topic from a collection of Amazon product reviews (Table 1). The anchor word "backpack" may seem strange. However, this dataset contains nothing about regular backpacks; thus, "backpack" is unique to camera bags. Bizarre, low-to-mid frequency words are often anchors because anchor words must be *unique* to a topic; intuitive or high-frequency words cannot be anchors if they have probability in *any other topic*.

If we instead ask users to give us representative words for this topic, we would expect combinations of words like "camera" and "bag." However, with single word anchors we must choose a single word to anchor each topic. Unfortunately, because these words might appear in multiple topics, individually they are not suitable as anchor words. The anchor word "camera" generates a general camera topic instead of camera bags, and the topic anchored by "bag" includes bags for diaper pails (Table 1).

Instead, we need to use sets of representative terms as an interpretable, parsimonious description of a topic. This section discusses strategies to build anchors from multiple words and the implications of using multiword anchors to recover topics. This extension not only makes anchors more interpretable but also enables users to manually construct effective anchors in interactive topic modeling settings.

## 2.1 Anchor Facets

We first need to turn words into an anchor. If we interpret the anchor algorithm geometrically, each row of $\boldsymbol{Q}$ represents a word as a point in $V$-dimensional space. We then model each point as a convex combination of anchor words to reconstruct the topic matrix $\mathbf{A}$ (Equation 1). Instead of individual anchor words (one anchor word per topic), we use anchor **facets**, or sets of words that describe a topic. The facets for each anchor form a new **pseudoword**, or an invented point in $V$-dimensional space (described in more detail in Section 2.2).

While these new points do not correspond to words in the vocabulary, we can express non-anchor words as convex combinations of pseudowords. To construct these pseudowords from their facets, we combine the co-occurrence profiles of the facets. These pseudowords then augment the original cooccurrence matrix $\boldsymbol{Q}$ with $K$ additional rows corresponding to synthetic pseudowords forming each of $K$ multiword anchors. We refer to this augmented matrix as $\boldsymbol{S}$. The rest of the anchor algorithm proceeds unmodified.

Our augmented matrix $\boldsymbol{S}$ is therefore a $(V + K) \times V$ matrix. As before, $V$ is the number of token types in the data and $K$ is the number of topics. The first $V$ rows of $\boldsymbol{S}$ correspond to the $V$ token types observed in the data, while the additional $K$ rows correspond to the pseudowords constructed from anchor facets. Each entry of $\boldsymbol{S}$ encodes conditional probabilities so that $S_{i,j}$ is equal to $p(w_i \mid w_j)$. For the additional $K$ rows, we invent a cooccurrence pattern that can effectively explain the other words' conditional probabilities.

## 2.2 Combining Facets into Pseudowords

We now describe more concretely how to combine an anchor facets to describe the cooccurrence pattern of our new pseudoword anchor. In tandem anchors, we create vector representations that combine the information from anchor facets. Our anchor facets are $\mathcal{G}_1 \ldots \mathcal{G}_K$, where $\mathcal{G}_k$ is a set of anchor facets which will form the $k$th pseudoword anchor. The pseudowords are $g_1 \ldots g_K$, where $g_k$ is the pseudoword from $\mathcal{G}_k$. These pseudowords form the new rows of $\boldsymbol{S}$. We give several candidates for combining anchors facets into a single multiword anchor; we compare their performance in Section 3.

**Vector Average** An obvious function for computing the central tendency is the vector average. For each anchor facet,

$$\boldsymbol{S}_{g_k,j} = \sum_{i \in \mathcal{G}_k} \frac{\boldsymbol{S}_{i,j}}{|\mathcal{G}_k|}, \tag{2}$$

where $|\mathcal{G}_k|$ is the cardinality of $\mathcal{G}_k$. Vector average makes the pseudoword $\boldsymbol{S}_{g_k,j}$ more central, which is intuitive but inconsistent with the interpretation from Arora et al. (2013) that anchors should be extreme points whose linear combinations explain more central words.

**Or-operator** An alternative approach is to consider a cooccurrence with *any* anchor facet in $\mathcal{G}_k$. For word $j$, we use De Morgan's laws to set

$$\boldsymbol{S}_{g_k,j} = 1 - \prod_{i \in \mathcal{G}_k} (1 - \boldsymbol{S}_{i,j}). \tag{3}$$

Unlike the average, which pulls the pseudoword inward, this or-operator pushes the word outward, increasing each of the dimensions. Increasing the volume of the simplex spanned by the anchors explains more words.

**Element-wise Min** Vector average and or-operator are both sensitive to outliers and cannot account for polysemous anchor facets. Returning to our previous example, both "camera" and "bag" are bad anchors for camera bags because they appear in documents discussing other products. However, if both "camera" and "bag" are anchor facets, we can look at an *intersection* of their contexts: words that appear with both. Using the intersection, the cooccurrence pattern of our anchor facet will only include terms relevant to camera bags.

Mathematically, this is an element-wise min operator,

$$\boldsymbol{S}_{g_k,j} = \min_{i \in \mathcal{G}_k} \boldsymbol{S}_{i,j}. \tag{4}$$

This construction, while perhaps not as simple as the previous two, is robust to words which have cooccurrences which are not unique to a single topic.

**Harmonic Mean** Leveraging the intuition that we should use a combination function which is both centralizing (like vector average) and ignores large outliers (like element-wise min), the final combination function is the element-wise harmonic mean. Thus, for each anchor facet

$$\boldsymbol{S}_{g_k,j} = \sum_{i \in \mathcal{G}_k} \left( \frac{\boldsymbol{S}_{i,j}^{-1}}{|\mathcal{G}_k|} \right)^{-1}. \tag{5}$$

Since the harmonic mean tends towards the lowest values in the set, it is not sensitive to large outliers, giving us robustness to polysemous words.

## 2.3 Finding Topics

After constructing the pseudowords of $S$ we then need to find the coefficients $C_{i,k}$ which describe each word in our vocabulary as a convex combination of the multiword anchors. Like standard anchor methods, we solve the following for each token type:

$$C_{i,\cdot}^* = \operatorname*{argmin}_{C_{i,\cdot}} D_{KL}\left(S_{i,\cdot} \, \middle\| \, \sum_{k=1}^{K} C_{i,k} S_{g_k,\cdot}\right). \tag{6}$$

Finally, we appeal to Bayes' rule, we recover the topic-word matrix $A$ from the coefficients of $C$.

## 3 High Water Mark for Tandem Anchors

Before addressing interactivity, we apply tandem anchors to real world data, but with anchors gleaned from metadata. Our purpose is twofold. First, we determine which combiner from Section 2.2 to use in our interactive experiments in Section 4 and second, we confirm that well-chosen tandem anchors can improve topics. In addition, we examine running time of tandem anchors and compare to traditional model-based interactive topic modeling techniques. We cannot assume that we will have metadata available to build tandem anchors, but we use them here because they provide a high water mark without the variance introduced by study participants.

## 3.1 Experimental Setup

We use the well-known 20 Newsgroups dataset (20NEWS) used in previous interactive topic modeling work: 18,846 usenet postings from 20 different newgroups in the early 1990s. We remove stopwords and words which appear in fewer than 100 documents or more than 1,500 documents.

To seed the tandem anchors, we use the titles of newsgroups. To build each multiword anchor facet, we split the title on word boundaries and expand any abbreviations or acronyms. For example, the newsgroup title 'comp.os.ms-windows.misc' becomes {"computer", "operating", "system", "microsoft", "windows", "miscellaneous"}. We do not fully specify the topic; the title gives some intuition, but the topic modeling algorithm must still recover the complete topic-word distributions. This is akin to knowing the names of the categories used but nothing else. Critically, the topic modeling algorithm has no knowledge of document-label relationships.

## 3.2 Experimental Results

Our first evaluation is a classification task to predict documents' newsgroup membership. We do not aim for state-of-the-art accuracy, but the experiment shows title-based tandem anchors yield topics closer to the underlying classes than Gram-Schmidt anchors. After randomly splitting the data into test and training sets we learn topics from the test data using both the title-based tandem anchors and the Gram-Schmidt single word anchors.[1] For multiword anchors, we use each of the proposed combiner functions from Section 2.2. The anchor algorithm only gives the topic-word distributions and not word-level topic assignments, we infer token-level topic assignments using Latent Dirichlet Allocation (Blei et al., 2003) with *fixed* topics discovered by the anchor method. We use our own implementation of Gibbs sampling with fixed topics and a symmetric document-topic Dirichlet prior with concentration $\alpha = .01$. Since the topics are fixed, this inference is very fast and can be parallelized on a per-document basis. We then train a hinge-loss linear classifier on the newsgroup labels using Vowpal Wabbit[2] with topic-word pairs as features. Finally, we infer topic assignments in the test data and evaluate the classification using those topic-word features. For both training and test, we exclude words outside the LDA vocabulary.

The topics created from multiword anchor facets are more accurate than Gram-Schmidt topics (Figure 1). This is true regardless of the combiner function. However, harmonic mean is more accurate than the other functions.[3]

Since 20NEWS has twenty classes, accuracy alone does not capture confusion between closely related newsgroups. For example, accuracy penalizes a classifier just as much for label-

---

[1] With fixed anchors and data the anchor algorithm is deterministic, so we use random splits instead of the standard train/test splits so that we can compute variance.

[2] http://hunch.net/~vw/

[3] Significant at $p < 0.01/4$ when using two-tailed t-tests with a Bonferroni correction. For each of our evaluations, we verify the normality of our data (D'Agostino and Pearson, 1973) and use two-tailed t-tests with Bonferroni correction to determine whether the differences between the different methods are significant.

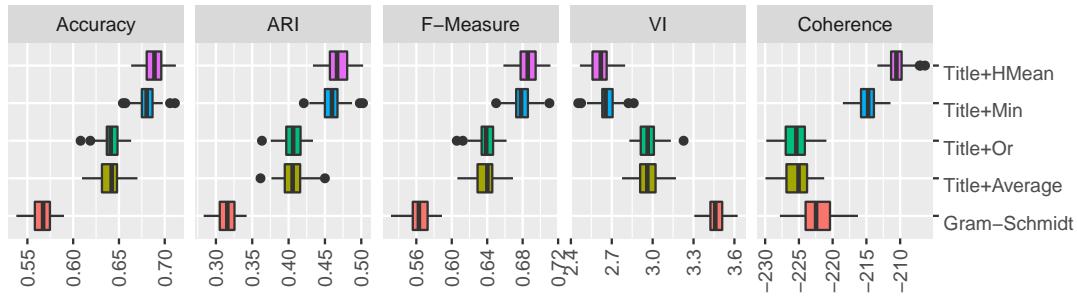

Figure 1: Using metadata can improve anchor-based topic models. For all metrics, the unsupervised Gram-Schmidt anchors do worse than creating anchors based on Newsgroup titles (for all metrics except VI, higher is better). For coherence, Gram-Schmidt does better than two functions for combining anchor words, but not the element-wise min or harmonic mean.

ing a document from 'rec.sport.baseball' with 'rec.sport.hockey' as with 'alt.atheism' despite the similarity between sports newsgroups. Consequently, after building a confusion matrix between the predicted and true classes, external clustering metrics reveal confusion between classes.

The first clustering metric is the adjusted Rand index (Yeung and Ruzzo, 2001), which is akin to accuracy for clustering, as it gives the percentage of correct pairing decisions from a reference clustering. Adjusted Rand index also accounts for chance groupings of documents. Next we use F-measure, which also considers pairwise groups, balancing the contribution of false negatives, but without the true negatives. Finally, we use variation of information. This metric measures the amount of information lost by switching from the gold standard labels to the predicted labels (Meilă, 2003). Since we are measuring the amount of information lost, lower variation of information is better.

Based on these clustering metrics, tandem anchors can yield superior topics to those created using single word anchors (Figure 1). As with accuracy, this is true regardless of which combination function we use. Furthermore, harmonic mean produces the least confusion between classes.[4]

The final evaluation is topic coherence by Newman et al. (2010), which measures whether the topics make sense, and correlates with human judgments of topic quality. Given $V$, the set of

the $n$ most probable words of a topic, coherence is

$$\sum_{v_1, v_2 \in V} log \frac{D(v_1, v_2) + \epsilon}{D(v_2)} \qquad (7)$$

where $D(v_1, v_2)$ is the co-document frequency of word types $v_1$ and $v_2$, and $D(v_2)$ is the document frequency of word type $v_2$. A smoothing parameter $\epsilon$ prevents zero logarithms.

Figure 1 also shows topic coherence. Although title-based anchor facets produce better classification features, topics from Gram-Schmidt anchors have better coherence than title-based anchors with the vector average or the or-operator. However, when using the harmonic mean combiner, title-based anchors produce the most human interpretable topics.[4]

Harmonic mean beats other combiner functions because it is robust to ambiguous or irrelevant term cooccurrences an anchor facet. Both the vector average and the or-operator are swayed by large outliers, making them sensitive to ambiguous terms in an anchor facet. Element-wise min also has this robustness, but harmonic mean is also able to better characterize anchor facets as it has more centralizing tendency than the min.

### 3.3 Runtime Considerations

Tandem anchors will enable users to direct topic inference to improve topic quality. However, for the algorithm to be interactive we must also consider runtime. Cook and Thomas (2005) argue that for interactive applications with user-initiated actions like ours the response time should be less than ten seconds. Longer waits can increase the cognitive load on the user and harm the user interaction.

---

[4]Significant at $p < 0.01/4$ when using two-tailed t-tests with a Bonferroni correction. For each of our evaluations, we verify the normality of our data (D'Agostino and Pearson, 1973) and use two-tailed t-tests with Bonferroni correction to determine whether the differences between the different methods are significant.

| Algorithm | #Doc | Runtime (seconds) |
|---|---|---|
| Tandem Anchors | 18828 | 2.13 |
| Fast ITM | 13284 | 24.8 |
| Utopian | 515 | 48.0 |

Table 2: Reported run times for various interactive topic modeling algorithms. Only tandem anchors is fast enough to be interactive. For Tandem Anchors, we ran on a single core of an AMD Phemon II X6 1090T processor. For Fast ITM, we use the time reported by Hu and Boyd-Graber (2012). For Utopian, we use the time published by Choo et al. (2013).

Fortunately, the runtime of tandem anchors is amenable to interactive topic modeling. Our run time results are summarized in Table 2. On 20NEWS, interactive updates take roughly two seconds. Furthermore, larger datasets typically have a sublinear increase in distinct word types, so we can expect to see similar run times, even on much larger datasets.

Compared to other interactive topic modeling algorithms, tandem anchors has a very attractive run time. For example, using an optimized version of the sampler for the Interactive Topic Model described by Hu and Boyd-Graber (2012), and the recommended 30 iterations of sampling, the Interactive Topic Model is well beyond our desired update time for interactive use and an order of magnitude slower than tandem anchors.

Another promising interactive topic modeling approach is Utopian (Choo et al., 2013), which uses non-negative factorization, albeit without the benefit of anchor words. Utopian is much slower than tandem anchors. Even on the small InfoVis-VAST dataset, Utopian takes nearly a minute to converge. Furthermore, since Utopian scales linearly with the size of the data, for moderately sized datasets such as 20NEWS, Utopian is infeasible for interactive topic modeling due to run time.

While each of these interactive topic modeling algorithms do achieve reasonable topics, only our algorithm fits the run time requirements for interactivity. Furthermore, since tandem anchors scales with the size of the vocabulary rather than the size of the data, this trend will only become more pronounced as we increase the amount of data.

## 4   Interactive Anchor Words

Given high quality anchor facets, the tandem anchor algorithm can produce high quality topic

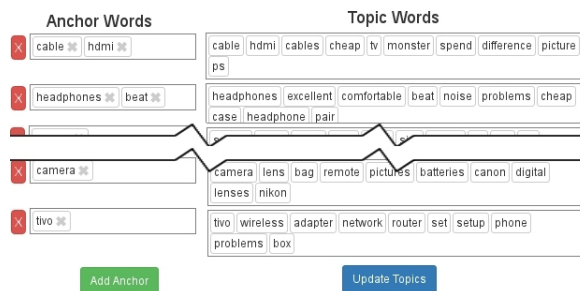

Figure 2: Interface for user study with multiword anchors applied to interactive topic modeling.

models (particularly when the harmonic mean combiner is used). Furthermore, the tandem anchor algorithm is fast enough to be interactive (as opposed to model-based approaches such as the Interactive Topic Model). We now turn our attention to our main experiment: tandem anchors applied to the problem of interactive topic modeling. We compare both single word and tandem anchors in our study. We do not include the Interactive Topic Model or Utopian, as their run times are too slow for our users.

### 4.1   Interface and User Study

To show that interactive tandem anchor words are fast, effective, and intuitive, we ask users to understand a dataset using the anchor word algorithm. For this user study, we recruit twenty participants drawn from a university student body. The student median age is twenty-two. Seven are female, and thirteen are male. None of the students had any prior familiarity with topic modeling or the 20NEWS dataset.

Each participant sees a simple user interface (Figure 2) with topic given as a row with two columns. The left column allows users to view and edit topics' anchor words; the right column lists the most probable words in each topic.[5] The user can remove an anchor word or drag words from the topic word lists (right column) to become an anchor word. Users can also add additional topics by clicking the "Add Anchor" to create additional anchors. If the user wants to add a word to a tandem anchor set that does not appear in the interface, they manually type the word (restricted to the model's vocabulary). When the user wants to see the updated topics for their newly refined anchors,

---

[5]While we use topics generated using harmonic mean for our final analysis, users were shown topics generated using the min combiner. However, this does not change our result.

they click "Update Topics".

We give each a participant a high level overview of topic modeling. We also describe common problems with topic models including intruding topic words, duplicate topics, and ambiguous topics. Users are instructed to use their best judgement to determine if topics are useful. The task is to edit the anchor words to improve the topics. We asked that users spend at least twenty minutes, but no more than thirty minutes. We repeat the task twice: once with tandem anchors, and once with single word anchors.[6]

## 4.2 Quantitative Results

We now validate our main result that for interactive topic modeling, tandem anchors yields better topics than single word anchors. Like our title-based experiments in Section 3, topics generated from users become features to train and test a classifier for the 20NEWS dataset. Based on our results with title-based anchors, we use the harmonic mean combiner in our analysis. As before, we report not only accuracy, but also multiple clustering metrics using the confusion matrix from the classification task. Finally, we report topic coherence.

Figure 3 summarizes the results of our quantitative evaluation. While we only compare user generated anchors in our analysis, we include the unsupervised Gram-Schmidt anchors as a baseline. Some of the data violate assumptions of normality. Therefore, we use Wilcoxon's signed-rank test (Wilcoxon, 1945) to determine if the differences between multiword anchors and single word anchors are significant.

Topics from user generated multiword anchors yield higher classification accuracy (Figure 3). Not only is our approach more scalable than the Interactive Topic Model, but we also achieve higher classification accuracy than Hu et al. (2014).[7] Tandem anchors also improve clustering metrics.[8]

While user selected tandem anchors produce better classification features than single word anchors, users selected single word anchors produce topics with similar topic coherence scores.[9]

---

[6]The order in which users complete these tasks is counterbalanced.

[7]However, the values are not strictly comparable, as Hu et al. (2014) use the standard chronological test/train fold, and we use random splits.

[8]Significant at $p < 0.01$ when using Wilcoxon's signed-rank test.

[9]The difference between coherence scores was *not* statis-

To understand this phenomenon, we use quality metrics (AlSumait et al., 2009) for ranking topics by their correspondence to genuine themes in the data. Significant topics are likely skewed towards a few related words, so we measure the distance of each topic-word distribution from the **uniform** distribution over words. Topics which are close to the underlying word distribution of the entire data are likely to be **vacuous**, so we also measure the distance of each topic-word distribution from the underlying word distribution. Finally, **background** topics are likely to appear in a wide range of documents, while meaningful topics will appear in a smaller subset of the data.

Figure 4 reports our topic significance findings. For all three significance metrics, multiword anchors produce more significant topics than single word anchors.[8]Topic coherence is based solely on the top $n$ words of a topic, while both accuracy and topic significance depend on the entire topic-word distributions. With single word anchors, topics with good coherence may still be too general. Tandem anchors enables users to produce topics with more specific word distributions which are better features for classification.

## 4.3 Qualitative Results

We examine the qualitative differences between how users select multiword anchor facets versus single word anchors. Table 3 gives examples of topics generated using different anchor strategies. In a follow-up survey with our users, 75% find it easier to affect individual changes in the topics using tandem anchors compared to single word anchors. Users who prefer editing multiword anchors over single word anchors often report that multiword anchors make it easier to merge similar topics into a single focused topic by combining anchors. For example, by combining multiple words related to Christianity, users were able to create a topic which is highly specific, and differentiated from general religion themes which included terms about Atheism and Judaism.

While users find that use tandem anchors is easier, only 55% of our users say that they prefer the final topics produced by tandem anchors compared to single word anchors. This is in harmony with our quantitative measurements of topic coherence, and may be the result of our stopping criteria: when users judged the topics to be useful.

---

tically significant using Wilcoxon's signed-rank test.

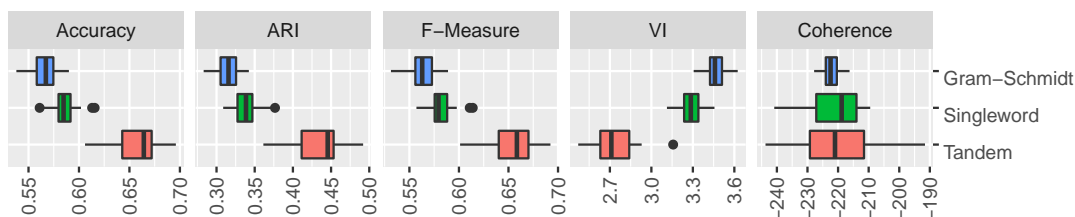

Figure 3: Classification accuracy and coherence using topic features gleaned from user provided multiword and single word anchors. Grahm-Schmidt anchors are provided as a baseline. For all metrics except VI, higher is better. Except for coherence, multiword anchors are best.

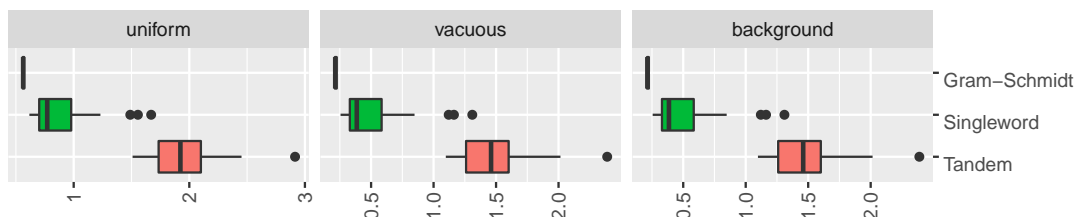

Figure 4: Topic significance for both single word and multiword anchors. In all cases higher is better. Multiword anchors produce topics which are more significant than single word anchors.

| Anchor | Top Words in Topic |
|---|---|
| **Automatic Gram Schmidt** | |
| love | love god evolution romans heard car |
| game | game games team hockey baseball heard |
| **Interactive Single-word** | |
| evolution | evolution theory science faith quote facts |
| religion | religion god government state jesus israel |
| baseball | baseball games players word teams car |
| hockey | hockey team play games season players |
| **Interactive Tandem** | |
| atheism god exists prove | god science evidence reason faith objective |
| christian jesus | jesus christian christ church bible christians |
| jew israel | israel jews jewish israeli state religion |
| baseball bat ball | hit baseball ball player games call |
| hockey nhl | team hockey player nhl win play |

Table 3: Comparison of topics generated for 20NEWS using various types of anchor words. Users are able to combine words to create more specific topics with tandem anchors.

However, 100% of our users feel that the topics created through interaction were better than those generated from Gram-Schmidt anchors. This was true regardless of whether we used tandem anchors or single word anchors.

Our participants also produce fewer topics when using multiword anchors. The mean difference between topics under single word anchors and multiple word anchors is 9.35. In follow up interviews, participants indicate that the easiest way to resolve an ambiguous topic with single word anchors was to create a new anchor for each of the ambiguous terms, thus explaining the proliferation of topics for single word anchors. In contrast, fixing an ambiguous tandem anchor is simple: users just add more terms to the anchor facet.

## 5 Conclusion

Tandem anchors extend the anchor words algorithm to allow multiple words to be combined into anchor facets. For interactive topic modeling, using anchor facets in place of single word anchors produces higher quality topic models and are more intuitive to use. Furthermore, our approach scales much better than existing interactive topic modeling techniques, allowing interactivity on large datasets for which interactivity was previous impossible.

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
