# Peer review of "Tandem Anchoring: a Multiword Anchor Approach for Interactive Topic Modeling"

_ACL 2017 — decision unknown_

[Official Review · Reviewer 1 · rating 4 · confidence 5]
soundness 3 · originality 3 · clarity 4 · impact 3 · substance 4 · appropriateness 5 · meaningful comparison 3 · presentation format Oral Presentation

- Strengths:

The paper offers a natural and useful extension to recent efforts in
interactive topic modeling, namely by allowing human annotators to provide
multiple "anchor words" to machine-induced topics. The paper is well-organized
and the combination of synthetic and user experiments make for a strong paper.

- Weaknesses:

The paper is fairly limited in scope in terms of the interactive topic model
approaches it compares against. I am willing to accept this, since they do make
reference to most of them and explain that these other approaches are not
necessarily fast enough for interactive experimentation or not conducive to the
types of interaction being considered with an "anchoring" interface. Some level
of empirical support for these claims would have been nice, though.

It would also have been nice to see experiments on more than one data set (20
newsgroups, which is now sort of beaten-to-death).

- General Discussion:

In general, this is a strong paper that appears to offer an incremental but
novel and practical contribution to interactive topic modeling. The authors
made the effort to vet several variants of the approach in simulated
experiments, and to conduct fairly exhaustive quantitative analyses of both
simulated and user experiments using a variety of metrics that measure
different facets of topic quality.

[Official Review · Reviewer 2 · rating 4 · confidence 5]
soundness 3 · originality 3 · clarity 5 · impact 3 · substance 3 · appropriateness 5 · meaningful comparison 3 · presentation format Oral Presentation

- Strengths:
Clear description of methods and evaluation
Successfully employs and interprets a variety of evaluations
Solid demonstration of practicality of technique in real-world interactive
topic modeling

- Weaknesses:
Missing related work on anchor words
Evaluation on 20 Newsgroups is not ideal
Theoretical contribution itself is small 

- General Discussion:
The authors propose a new method of interactive user specification of topics
called Tandem Anchors. The approach leverages the anchor words algorithm, a
matrix-factorization approach to learning topic models, by replacing the
individual anchors inferred from the Gram-Schmidt algorithm with constructed
anchor pseudowords created by combining the sparse vector representations of
multiple words that for a topic facet. The authors determine that the use of a
harmonic mean function to construct pseudowords is optimal by demonstrating
that classification accuracy of document-topic distribution vectors using these
anchors produces the most improvement over Gram-Schmidt. They also demonstrate
that their work is faster than existing interactive methods, allowing
interactive iteration, and show in a user study that the multiword anchors are
easier and more effective for users.

Generally, I like this contribution a lot: it is a straightforward modification
of an existing algorithm that actually produces a sizable benefit in an
interactive setting. I appreciated the authors’ efforts to evaluate their
method on a variety of scales. While I think the technical contribution in
itself is relatively small (a strategy to assemble pseudowords based on topic
facets) the thoroughness of the evaluation merited having it be a full paper
instead of a short paper. It would have been nice to see more ideas as to how
to build these facets in the absence of convenient sources like category titles
in 20 Newsgroups or when initializing a topic model for interactive learning.

One frustration I had with this paper is that I find evaluation on 20
Newsgroups to not be great for topic modeling: the documents are widely
different lengths, preprocessing matters a lot, users have trouble making sense
of many of the messages, and naive bag-of-words models beat topic models by a
substantial margin. Classification tasks are useful shorthand for how well a
topic model corresponds to meaningful distinctions in the text by topic; a task
like classifying news articles by section or reviews by the class of the
subject of the review might be more appropriate. It would also have been nice
to see a use case that better appealed to a common expressed application of
topic models, which is the exploration of a corpus.

There were a number of comparisons I think were missing, as the paper contains
little reference to work since the original proposal of the anchor word model.
In addition to comparing against standard Gram-Schmidt, it would have been good
to see the method from Lee et. al. (2014), “Low-dimensional Embeddings for
Interpretable Anchor-based Topic Inference”. I also would have liked to have
seen references to Nguyen et. al. (2013), “Evaluating Regularized Anchor
Words” and Nguyen et. al. (2015) “Is Your Anchor Going Up or Down? Fast and
Accurate Supervised Topic Models”, both of which provide useful insights into
the anchor selection process.

I had some smaller notes:
- 164: …entire dataset
- 164-166: I’m not quite sure what you mean here. I think you are claiming
that it takes too long to do one pass? My assumption would have been you would
use only a subset of the data to retrain the model instead of a full sweep, so
it would be good to clarify what you mean.
- 261&272: any reason you did not consider the and operator or element-wise
max? They seem to correspond to the ideas of union and intersection from the or
operator and element-wise min, and it wasn’t clear to me why the ones you
chose were better options.
- 337: Usenet should be capitalized
- 338-340: Why fewer than 100 (as that is a pretty aggressive boundary)? Also,
did you remove headers, footers, and/or quotes from the messages?
- 436-440: I would have liked to see a bit more explanation of what this tells
us about confusion.
- 692: using tandem anchors

Overall, I think this paper is a meaningful contribution to interactive topic
modeling that I would like to see available for people outside the machine
learning community to investigate, classify, and test hypotheses about their
corpora.

POST-RESPONSE: I appreciate the thoughtful responses of the authors to my
questions. I would maintain that for some of the complimentary related work
that it's useful to compare to non-interactive work, even if it does something
different.